# Please Pass the Translanguaging: The Dinner Table Experience in the Lives of Newcomer Canadian Deaf Youth and Their Families

**Joanne Catherine Weber [1,\*], Chelsea Temple Jones [2] and Abneet Atwal [2]**

1   Faculty of Education, University of Alberta, Edmonton, AL T6G 2E9, Canada
2   Department of Child and Youth Studies, Faculty of Social Sciences, Brock University,
    St. Catharines, ON L2S 3A1, Canada; cjones@brocku.ca (C.T.J.); aatwal@brocku.ca (A.A.)
\*   Correspondence: jweber1@ualberta.ca

**Abstract:** While translanguaging occurs in the homes of deaf people and their hearing family members who do not sign or possess limited signing skills, in this article we argue that translanguaging alone does not explain the complex, domestic-sphere language experiences of three young, newcomer artists in Saskatchewan, Canada. We frame our inquiry around the "dinner table experience" phenomenon, wherein deaf family members receive partial or little access to conversational exchanges. At the dinner table, which is both a literal setting and a metaphor for exclusion experienced by deaf people in audiocentric cultures, many deaf family members report feeling loved yet disconnected. However, translanguaging serves to expand linguistic repertoires among hearing and deaf interlocutors amidst the dinner table experience. We draw from three interviews with deaf youth who describe the dinner table experience through both dialogue and art making, including descriptions of ways in which communication is facilitated or not facilitated, thereby highlighting available and unavailable translanguaging practices in the domestic sphere. The interview data suggest that the dinner table experience is a significant setting for translanguaging, and that promoting accessible and equitable translanguaging practices in the home remains a significant challenge, especially when combined with newcomer lived experience that does not always match current descriptions of translanguaging. We posit that translanguaging is a necessary practice among hearing and deaf persons at the table that can and should be expanded to consider the intersectional experiences of communicators in this literal and metaphorical setting.

**Keywords:** deaf; translanguaging; dinner table experience; arts-based action research; identity

## 1. Introduction

This paper critically explores the complex translanguaging experiences of three deaf newcomer youth in Saskatchewan, Canada, and their families. In Canada, the term "newcomer" espouses refugees and immigrants who have resided in Canada for a short time (usually less than five years) to denote the variations in life circumstances that require people to relocate from one country to another (Government of Canada 2022). Deaf newcomers often have grown up with limited or no access to a language within their home community and may not have had opportunities to learn sign language in their countries of origin (Allard and Wedin 2017; Holmström et al. 2021). Their positioning requires an axiomatic analysis of race, gender, age, and deaf culture given that their lives intersect with normalizing processes that are interconnected and collusive and, as we demonstrate below, include but cannot be entirely explained through a lens of translanguaging.

Translanguaging often occurs in the homes of deaf people and their hearing family members who do not sign or possess limited signing skills. Translanguaging practices in the homes of immigrant deaf Canadians are complex, spanning beyond the use of multiple languages to encompass unique linguistic repertoires and cultural dynamics in

need of further study (Holmström et al. 2021). For this reason, we focus primarily on translanguaging instinct (Wei 2018) in a deaf context that may evolve alongside other intersectional factors, including race and gender. Wei (2018) defines the translanguaging instinct as the urge or drive to extend one's repertoire beyond strictly defined linguistic cues to negotiate meaning and establish effective communication. Wei (2018) suggests that human beings rely on multiple resources, whether they be sensory, modal, cognitive, or semiotic, to facilitate understanding between interlocutors. The translanguaging instinct enables interlocutors to identify gaps in meaning, make interferences, and draw upon multiple resources to interact with others (Wei 2018). The authors emphasize that the following paper is not a linguistic ethnography in which utterances are analyzed but a critical examination of how the translanguaging instinct is employed at the dinner table. Our purpose is to elaborate on how the interchanges between language learning and linguistic, social, and cultural capital shape the translanguaging experiences of deaf newcomer Canadians.

In this article, we consider translanguaging as it emerges in the three participants' respective domestic spheres using the "dinner table experience" phenomenon, a unique translanguaging experience wherein deaf family members receive partial or little access to conversational exchanges during family or community gatherings (Hall et al. 2018). Typically referred to as "dinner table syndrome", this experience is shorthand for a phenomenon that is both literal and metaphorical: while experiences of exclusion may, indeed, happen around an actual dinner table, the "dinner table experience" is often conceived of more broadly as the ideological experience of missing out on overlapping conversations in a wide variety of contexts (Meek 2020). This experience is especially felt by deaf people for whom most of their kin are non-native users of sign language (Mitchell and Karchmer 2005) and often render signed language as subordinate to spoken language. Amidst this experience—or, one might say, *at the dinner table*—many deaf family members report feeling loved yet disconnected (Meek 2020). The anxiety and tensions inherent in learning the dominant language and maintaining their own minority language (signed language) often result in deaf acculturative stress (Aldalur et al. 2021). At the same time, translanguaging serves to expand linguistic repertoires among hearing and deaf interlocutors by surfacing different linguistic, social, and cultural practices between deaf young adult newcomers and their hearing family members (Iturriaga and Young 2022).

We draw from three interviews with young adult deaf newcomer artists, as well as their public-facing performances and installations, to describe their encounters with the dinner table experience both as the literal experience of translanguaging while dining with their families and as the broader experience of exclusions from audiocentric cultures. These experiences include ways in which communication is facilitated or not facilitated, thereby highlighting available translanguaging practices in the domestic and public spheres. With a focus on interactions in the private sphere, the interview data suggest that promoting accessible and equitable translanguaging practices in the home remains a significant challenge, especially when combined with newcomer lived experience. Ultimately, we posit that the dinner table experience may be one of the most highly contested spaces in which identity, language ideologies, investments in language learning, and linguistic, social, and cultural capital shape the translanguaging experiences of deaf newcomer Canadians.

## 2. Translanguaging and Capital in a Deaf Context

Translanguaging draws upon the plurilinguistic repertoires of individual interlocutors (Canagarajah 2013). Partial mastery of a language can enable the development of further linguistic competencies in settings where multiple languages are in use. Translanguaging activates existing plurilingual competencies to achieve an outcome desired by all interlocutors. Translanguaging is derived from a single set of linguistic skills which in turn support the use of multiple languages to co-construct meaning (Wei and García 2022). While translanguaging recognizes that languages are unbounded and heterogeneous (Garcia et al. 2015), translanguaging practices can be constrained or defined according to language ideologies,

language-learning commitments, and cultural practices, meaning that domestic-sphere contexts can complexify translanguaging (Canagarajah 2013; Wei and García 2022). For example, some participants do possess some knowledge of their families' languages either in print form or as a spoken language, which was often constrained by their hearing and speaking abilities. In contrast, in gatherings with deaf people and hearing people who sign with reasonable fluency, the deaf youth enjoyed the free and easy exchange of information, ideas, and knowledge of current and past events with each other and other deaf individuals who were not part of their families of origin. Within academic contexts, translanguaging as a pedagogical maneuver is gaining traction in university and K–12 classrooms (Canagarajah 2013; Lin 2019). In these other contexts such as the classroom, theatre rehearsals, and restaurant meals attended only by deaf people who sign, the deaf youth were able to construct new knowledge and develop expanded understandings facilitated by their use of sign language, which enabled them to expand their linguistic repertoire in other languages and to devise a sophisticated, multilayered theatre performance. Translanguaging for deaf individuals, therefore, remains precarious as complete and sustained access to any language (spoken or sign) is not guaranteed (Snoddon and Weber 2021).

Translanguaging theories are embedded within the sociocultural frame advanced by Vygotsky, in which social interactions constitute the bedrock of the construction of knowledge, as well as understanding of one's self, others, and the world beyond the self (Lin 2019). Within deaf populations, translanguaging may involve the use of additional resources that are nonverbal, such as gestures, material resources, and semiotic resources in order to arrive at agreed-upon meanings between individuals who possess differing competencies across languages (Kusters 2019; Kusters et al. 2017). As we demonstrate below, the process of meaning making afforded by translanguaging in contexts other than the dinner table included reference to physical resources such as props, furniture, puppets, and physical movement including clowning techniques that emerge in the participants' engagement with public-facing artistry. The multiplicity of physical and semiotic (such as signage) resources contributed to their ability to engage in translanguaging with each other and the audience. This is known as a form of trans-semioticization in which gestures, facial expressions, and visual images are used as part of the translanguaging activity between interlocutors (Lin 2019). Therefore, translanguaging theories embrace fluidity and the construction of knowledge in a dialogic space (Lin 2019; Freire 2000).

It is possible to imagine the "dinner table experience" as a literal and metaphorical dialogic space. However, the nature of the dinner table experience suggests that translanguaging experiences of deaf newcomer Canadians are not only characterised according to the availability of expanded linguistic and semiotic resources, but are complicated by the available social, cultural, and linguistic capital of parents and their deaf children, abstracted and situated language ideologies, social–cultural beliefs, and identities which often do not acknowledge deaf members' sensory orientations and their unequal access to semiotic resources (De Meulder et al. 2019a; Murray et al. 2020; Snoddon and Underwood 2014). Social capital is defined by Bourdieu as the social value accrued through participation in established institutions such as the obtaining of diplomas, degrees, and memberships that translate into real social, economic, and agentic power (Bourdieu and Thompson 1991). Cultural capital comprises the social assets of a person with respect to education, style of speech, forms of dress, and participation in religious, cultural, and social traditions that ultimately provide social status and power to the individual as embedded within their community (Bourdieu 1986). Linguistic capital is best understood as the linguistic repertoires and competence required to participate in spaces legitimized by social, cultural, and economic institutions. In deaf education, "so much emotion, energy and ideology has been tied up in debates about which languages and modalities are most effective as a medium of education" (O'Brien 2021, p. 64). The linguistic capital of refugee and immigrant deaf persons with limited or no access to a language and who may have not experienced much schooling, is often diminished (Holmström et al. 2021).

Translanguaging theories do, however, lend support to the analysis of racial hierarchies and language ideologies. Therefore, translanguaging supports the legitimacy of plurilingual interlocutors in participants' language usage and language styles and registers (Flores and Garcia 2017). Further, collaboration is at the heart of translanguaging where relationships are forged, meanings are established, and problems are solved (Swain and Watanabe 2013). However, though translanguaging can level the playing field by not reinforcing binarized categories that promote deficit perspectives and language hierarchies (Lin 2019), its practices do not eradicate communication problems related to racism, classism, or other significant identarian factors with impacts on translanguaging experiences, including those that emerge at the dinner table for the three participants involved in this research. The deaf newcomer youth in this study demonstrate complex translanguaging practices activated through the development of the devised theatre performances and have also established themselves as deaf persons with histories, memories, and experiences of growing up in Bangladesh, Pakistan, and Syria and retaining many aspects of their cultural, religious, and social practices that are reflective of their family lives. These people occupy unique, intersectional positionings that are contextually contingent and not entirely resolvable through a lens of translanguaging.

In considering the nuanced translanguaging context of the dinner table for deaf newcomer Canadians, the interrelationships between translanguaging in deaf and hearing populations, deaf and hearing identity, deaf and hearing language ideologies, deaf and hearing investments, and social–cultural capital possessed by newcomer Canadians (deaf and hearing) remain unexplored and undertheorized. In other words, although it is an effective framework for understanding dinner table experiences, translanguaging alone does not adequately explain what happens among these young, deaf newcomers and hearing interlocutors at the literal and metaphorical dinner table. Using a comprehensive investment model proposed by Darvin and Norton (2015), which locates identity and investment within the social turn of applied linguistics, this paper critically explores and expands upon the translanguaging experiences of three young deaf newcomer Canadian artists, drawing on both their storytelling and their artistry.

### 3. Epistemology/Ontology/Axiology

The dinner table experience is often a muster point for family members who return from their own daily contexts and routines with stories to tell about themselves and the community at large, often with the purpose of strengthening family and ethnic ties and sociocultural forms of capital. However, as Meek (2020) reports in their wider articulation of the dinner table as both literal and metaphorical, dialogue at the literal dinner table is characterised by rapid turn taking, which makes it very difficult for deaf family members to comprehend what is spoken—a pattern that repeats in other social contexts beyond the dinner table. Hearing family members may talk over each other in a bid to interject, control the conversation, or provide a different emphasis on a particular aspect of the conversation. Asking for clarification often results in delayed explanations, incomplete relay of information, or annoyance on the part of hearing interlocutors. These common experiences suggest an axiological conflict, in which values associated with biosocial powers (Friedner 2010) afforded by hearing (as in spoken language) and by seeing (as in sign language) inform "what we are", "what we know", and "how we learn" (Skyer 2021). Despite Meek's (2020) report that deaf children in hearing families feel loved, their attempts to become fully integrated *at the dinner table* are often curtailed by axiological assumptions held by hearing parents and hearing siblings; that is, deaf people's perceptions of the world are ordered according to hearing orientations, no matter how faulty they may be. Reports of successful translanguaging experiences of deaf people suggest a set of negotiated axiological commitments in which the visu-centric needs of deaf people are balanced with competencies of hearing people who may be not as visu-centric or more audiocentric (Iturriaga and Young 2022). In these incidents, deaf people can build multilingual repertoires and use a wider range of semiotic resources to meet the challenges associated with an

incomplete spoken language repertoire with hearing people who, at the same time, possess limited facility with sign language. In gatherings with others, deaf members are bringing their translanguaging experiences, which may have occurred in a variety of other settings such as the school classroom, offices, and community clinic settings. In short, deaf family members may have already experienced successful incidences of translanguaging in other contexts that transcend simplistic deaf/hearing binaries and speak to the complexities of translanguaging beyond the use of different languages.

The dinner table experience is a metaphor for unresolved axiological conflict, and for uneven displays of biosocial power and its accompanying values. What is brought to the dinner table is often abstracted into the form of stories from outside the family home, anecdotes, humorous jokes heard and shared, or reinforcement of cultural, social, and political ideologies. Access to current news and events and to incidental forms of knowledge is a prominent feature of dinner table conversations (Meek 2020). The dinner table is a forum for encapsulated histories; stories that have happened outside the family home or in other rooms; references, direct or oblique, to interrelationships between known persons; and a limited range of material and semiotic resources such as food, plates, cutlery, serving utensils, seating arrangements, windows, doors, and lights. The pace of rapid dinner conversation and its transitory and shifting nature does not allow for the location and transport of additional linguistic, material, and semiotic resources to be easily reinserted into a conversation that has long moved on to another topic. We suggest that axiological commitments need to be examined to understand translanguaging practices between deaf and hearing interlocutors at the dinner table. With respect to translanguaging, these commitments are understood as investments in multiple-language learning (Darvin and Norton 2015). These make for a critical translanguaging space (Hamman 2018) in which micro- and macro-level power flows highlight the investments exercised by deaf and hearing interlocutors at the dinner table.

## 4. Methodology

### 4.1. Model of Investment

In this paper, we modify Darvin and Norton's (2015) comprehensive model of investments in language learning, which considers structure and agency as they occur at the intersection of identity, ideology, and capital. Originally applied to language learning in students who are learning a second language, this model points to "increasingly invisible" and complex power structures and their significant impact on identity and language learning (p. 51). Their case study leads us to our primary inquiry: How is translanguaging between deaf and hearing interlocutors at the dinner table affected by the interrelationships between the hidden and known multiple investments with respect to identity, ideology, and capital? By exploring the investments with respect to commitments to development of different types of identity, ideology, and capital (linguistic, cultural, and social), we can better understand the opportunities and potential for translanguaging among hearing and deaf people who are newcomer Canadians. Notably, Norton (2013) has argued that investments in learning a language result in increased cultural, social, and linguistic capital. Investment in identities can also position the learner in multiple and uneven conditions of power, which can lead to varying learning outcomes. These investments can highlight the complexity of identities that can shift across space and time and are produced and reproduced in multiple interactions. Shifts in identity involve negotiations of power in different fields and identity formation can be complex, contradictory, and fluid (Norton 2013). Language ideologies address the social and cultural values attached to the use of a language and its potential for power and agency within multiple worlds. For instance, English is used as a language of business, multinational transactions, and education (Canagarajah 2013).

Translanguaging at the material and metaphorical dinner table as it is positioned as a part of the "dinner table experience" is complicated by three primary interrelationships between hidden and known investments held by deaf and hearing interlocutors, particularly in newcomer Canadian families with deaf members. Within a critical translanguaging

space, we use an investment model perspective to examine the investments of identity, capital, and language ideology surfacing at the dinner table. Within cultural and social capital realms, investment occurs via the available affordances and perceived benefits of learning multiple languages which are largely shaped by available audiocentric- or visu-centric-derived forms of biosocial power (Skyer 2021). The tension inherent in acculturation into Canadian society and maintaining ethnic, social, cultural, linguistic, and religious beliefs results in acculturative stress which affects deaf family members differently and affects their translanguaging practices at the dinner table. For instance, family members are aware through their social interactions within their communities and Canadian society that the English language carries the most weight in terms of potential social and cultural capital to be accrued in their adopted country (Bourdieu and Thompson 1991; Canagarajah 2013). At the same time, ideology establishes systematic patterns of control through schools, workplaces, and higher education, which shapes language-learning investments and determines which language is worthy of more time and study. Mauldin (2016) reports on the proliferation of cochlear implant clinics designed to promote speech and listening skills in deaf children and the ideological conflicts over the use of sign language in addition to spoken language. Such ideological commitments shape the habitus of language users and position them to act and think a certain way and, at the same time, grant them agency to change their multiple contexts (O'Brien 2021).

Finally, multiple identities shape language-learning investments through the need to belong to multiple communities and to cultivate multiple cultural affiliations. Graif (2018) reports that identity is often under erasure in families with strong ethnic and cultural affiliations. Being deaf is equated with nonbeing and personal histories, contexts, and preoccupations are often rendered as unintelligible or nonexistent (Graif 2018). Being a functioning member of an ethnic community, if it means "passing" as one who can hear and speak (albeit in a limited sense), is more highly desirable than to have a deaf person in the family who uses sign language which is unintelligible to hearing family members and the surrounding hearing culture. Thus, identity investments contribute to investments in learning multiple languages (Darvin and Norton 2015). With these facets—identity, ideology, and capital—in mind, our research questions, and several sub-questions, take shape in ways that are modelled after Darvin and Norton's (2015) case study of two language learners and their investments at the intersection of identity, ideology, and capital:

1.  How invested are hearing and deaf interlocutors in their present and imagined identities? In what ways are they positioned by others, and how do they, in turn, position interlocutors in ways that grant or refuse power? How can hearing and deaf interlocutors gain from or resist these positions?
2.  What do nondeaf and deaf interlocutors perceive as benefits of investment, and how can the capital they possess serve as an affordance for learning?
3.  What systemic patterns of control (policies, codes, institutions) make it difficult to invest and acquire certain capital? How have prevailing language ideologies structured learners' habitus and predisposed them to certain ways of thinking?

*4.2. Arts-Based Action Research*

To address the above research questions, we turn to deaf youth newcomers who are actively involved in creating public art about their experiences. Art making is a mode of experience sharing such that participants' artistry intermingled with their interview responses and led to new, interpretive responses to our research questions. Leavy suggests that arts-based researchers use the notion of shape to understand how the artistic "form shapes the content and how audiences receive that content" (Leavy 2017, p. 2). Here, research data are presented in different shapes. In this study, shapes such as anecdotal storytelling during the interviews, the development of a scene about the dinner table in a theatre performance called *The Madcap Misadventures of Mustafa (2022)*, and the arts installation titled *From Deaf Shame to Deaf Same* (Bamford 2022) in which one section features the dinner table, all present the dinner table in multiple shapes that shift through participants' telling of experiences. In this data

collection, we begin to explore an "aesthetic intersubjective paradigm" (Chilton et al. 2015) in which the dinner table is more than a physical artefact, something that is often out of reach to the casual observer: "Who ever said the table was functional, intact, and inclusive? We can imagine the table as untidy, wobbly, and even overturned" (Jones et al. 2023). Here, sensory, emotional, perceptual, kinaesthetic, and imaginal forms of knowledge come to the fore in the art making by the artists in this study. The aesthetic intersubjective paradigm allows for access to multiple dimensions of the human experience not easily accessible through usual research approaches (Chilton et al. 2015). Intersubjective realities created and co-constructed by the three deaf youth research participants through theatre performance and their influence on the art installation give rise to knowledge that is often elusive and not readily apparent to those with audiocentric privilege. There is a therapeutic aspect to this research, however unintended by the research team. While it has been reported that deaf family members experience feelings of being disconnected from the family at the dinner table, the deaf youth in this study were able to use the arts as a form of knowledge creation and resistance to their commonly held experiences of being left behind. Moreover, they were able to influence younger deaf children and youth to develop their own art about the dinner table experience.

In this study, arts-based research practices include data generation through American Sign Language (ASL) storytelling about the dinner table experience in interviews, performances, and visual arts. Researchers interviewed the deaf youth and included questions about the dinner table as a metaphor for familial and community gatherings. The deaf youth artists, in collaboration with a Cirque du Soleil clown (Mooky McGuinty), decided to carry this topic into their artistic practice within the Deaf Crows Collective and in their interactions with deaf youth who included that topic within their arts installation, *From Deaf Shame to Deaf Same* (Bamford 2022). At this juncture, the research team was not involved in the cocreation of the dramatic presentation of the dinner table scene and the art installation. These artistic pieces enabled the researchers to undertake a performance study of the scene featuring the dinner table experience in *Madcap Misadventures of Mustafa* (Deaf Crows Collective 2022), which premiered on 17 June 2022. Finally, the researchers were able to view the visual arts installation and study the section on the dinner table as created by the deaf youth supported in the partnership between the Deaf Crows Collective, a school board, and the artist in residence, Chrystene Ells. The multiple movements between the ASL storytelling in the interviews, the performance piece, and the visual arts installation allow for a study of the investments undertaken by deaf and hearing interlocutors at the dinner table and beyond the dinner table. Since the arts-based data emanated from the participants' life experiences as shared in their storytelling, performance, and visual art, we position this inquiry as arts-based action research. Arts-based action research emphasises "experiential knowledge, artistic practices and an emergent, open-ended evolving inquiry process" (Leavy 2017, p. 194), and allows the researcher to interpret interactions in the light of investments of all present at the dinner table and of the artivists themselves, as we do below.

The connection between translanguaging and arts-based research may not be readily apparent. We argue that while translanguaging opens a space for including multiple material and semiotic resources during the process of constructing meaning between interlocutors, arts-based action research illuminates multimodal practices which are also inherent in studying the axiological commitments that need to be examined to deepen our understandings of translanguaging (Pennycook 2018). Language is multimodal, and is the result of interactions with humans, material and semiotic resources, and nonhuman actors (Pennycook 2018). At the same time, multimodality is also the provenance of arts-based research.

## 5. Researcher Positionality

The lead researcher is white, deaf, Canadian-born, fluent in ASL, a playwright, and involved in the deaf theatre scene in Canada. Importantly, Author 1 also resides in the

same city as the participants and maintains regular contact with them, making it possible to engage in processual and ongoing consent to participate in this research amidst a two-month interview schedule surrounded by a public performance and art installation. Author 2 is white, hearing, and involved in arts-based research and Critical Disability Studies. Also born in Canada, this team member is based in the province of Ontario and does not have regular contact with participants. The third member of the team is a South Asian Canadian, a child of immigrants, hearing, and a doctoral candidate whose research is grounded in Critical Disability Studies and DisCrit. This team member is based in Ontario and maintains regular contact with participants.

## 6. Methods

### 6.1. Semistructured ASL Interviews

This study is premised on interviews with the artists involved in the arts-based action research described above. Between December 2021 and January 2022, researchers interviewed three young deaf newcomer artists. These semistructured interviews were conducted through Zoom video conferencing software and lasted approximately one hour. Notably, the interviews were conducted in ASL and later converted into English transcripts for analysis by all authors, two of whom are not fluent in ASL. Conducting the interviews in ASL was a significant step in the methodology because it reflected a communication mode most accessible to the interviewees. Interview transcripts were analysed in two ways. First, Author 1 reviewed the video transcripts (in ASL) and made note of themes and moments in the conversations that seemed applicable to our earlier questions. Authors 2 and 3 reviewed the written, English transcripts and coded for themes using NVivo software. The authors compared their themes and made decisions about which narrative moments to elevate in the data analysis below. The research team coded their data and established themes related to the following categories: ideological beliefs about sign language held by family members, investments in language learning and translanguaging practices, externally imposed limitations on the use of linguistic repertoires, acquisition of cultural capital through performing arts, the use of additional resources in aiding communication (phone, paper), and spaces established for one language and not another. The interviews were translated by the interviewer (deaf) and their veracity was checked by a follow-up session with the youth performers in December 2022.

### 6.2. Performance and Installation Art

The newcomer Canadian deaf youth who participated in this study also engaged in their own artistry before, during, and after the interviews, which contributed to the arts-based action methods embedded in this research. Since 2016, these young people have been engaged in making art in the hope of making their own language, identity, and inner worlds visible to their families both within and outside of research contexts (Weber 2018, 2021a, 2021b). Being from Bangladesh, Pakistan, and Syria, they are youth performers and artists from the Deaf Crows Collective, and have engaged in "artivism", a confluence of art making, activism, and community engagement (Leduc 2016) with hearing and deaf interlocutors. As deaf community members, they have created art that is designed to educate the public, strengthen the deaf community, and assert their identity as deaf people in addition to their strong affiliations with their cultural identities as promoted within their families, and the right to use ASL in multiple contexts, including their own homes. They also express a cultural and social commitment to their families' embracing of the Muslim faith and traditions. The theme of the dinner table runs throughout the anecdotal storytelling in interviews with the deaf performers; a theatre performance, *Madcap Misadventures of Mustafa* (Deaf Crows Collective 2022); and their ongoing affiliations with deaf youth in a high school program resulting in a collaborative arts installation, *From Deaf Shame to Deaf Same* (Bamford 2022).

*6.3. Ethical Considerations*

Research ethics permission for this study was obtained in May 2021 from the University of Alberta and again in December 2022 from both the University of Alberta and Brock University to allow for the revealing of the identities of the three participants in the interview process. In addition, some of their material below emerged from public performances occurring at the Artesian Theatre, Regina, Saskatchewan on 17, 18, and 19 June 2022 of which the interviewees were a part. All actors are referred to by their actual names in the interview data and the public performances, as publicised in the On Cue Performance Hub website and in the program distributed to theatre patrons (On Cue Performance Hub 2022). All references to their creative work include actual names of performers, as per their preferences; upon learning about this article and their roles in it, all participants indicated that they preferred to be identified by their real names. With this preference in mind, Author 1 met with each participant again in December 2022 to share the content of the manuscript and to verify the veracity of the interviews. Descriptions of the interviewees, below, were written with permission and are based on information obtained both from our research interviews and through public performances and art installations that, in some instances, include biographical details.

**7. Performer Data**

*7.1. Fatima Nafisa*

Fatima Nafisa was introduced to ASL and deaf culture as a student in a high school program in Regina, Saskatchewan. She had a major role in the flagship production *Deaf Crows* which highlighted the impact of language deprivation on her social, emotional, and academic development (Deaf Crows Collective 2016). The following year, she participated in the development of the *Deaf Forest*, an arts installation which explored the affordances of the "hearing forest", the "hard of hearing forest", and the "deaf forest" (Deaf Crows Collective 2017). She then created a performance piece which highlighted the tension between her and her family members over her use of sign language and her newly developed identity (Deaf Crows Collective 2018). In 2020, she created a solo piece commenting on the fractured relationship between her and her mother, titled "I never saw", in which she performs an elegiac ASL piece mourning their lack of communication due to her mother's insistence on using spoken English instead of sign language (Nafisa 2021). In December 2021, Fatima, with Mustafa Alabssi, Kainat Wahid, and Shayla Rae Tanner, an Indigenous deaf woman, created a video, *Deaf to Deaf: Research and Stories* for the Inclusive Early Childhood Service System project, outlining their experiences with school and family (Alabssi et al. 2022; Underwood and Snoddon 2021).

*7.2. Kainat Wahid*

Kainat Wahid became engaged in the production of the *Deaf Forest* installation, lending her skills and model building to the construction of large wall hangings, a fabric tree, masks, and signage (Deaf Crows Collective 2017). In a 2018 production called *Apple Time*, she told the story of how she had been pursued by a stalker in Pakistan and the resulting anxiety that kept her awake at night despite the move to Canada (Deaf Crows Collective 2018). The story emphasised how the acquisition of deaf friends who, through their use of sign language and membership in the deaf community, contribute to her feeling safe again. Kainat also collaborated with Fatima and Mustafa in the creation of the 2021 *Deaf to Deaf* video, again sharing her story of attending school in Pakistan and being made to feel stupid and ineffectual (Alabssi et al. 2022).

*7.3. Mustafa Alabssi*

Mustafa Alabssi is a Syrian newcomer to Canada who, at age 17, immigrated to Canada. Upon entering the deaf and hard of hearing resource room a few months later, he began learning ASL and met Fatima and Kainat in the same program. He participated in the creation of the *Deaf Forest* art exhibition, using his skills in carpentry and welding. With

the support of Mooky McGuinty, a Cirque du Soleil clown, he developed a clown act for *Apple Time*, encapsulating his schooling experiences in Syria, the outbreak of the war, time in a refugee camp, and his arrival in Canada where he learned ASL. He also landed a role in the *Black Summer* series with Netflix in the summer of 2018. He communicates with his family using a mixture of homemade signs, gestures, and some Arabic signs. He learned ASL upon arrival in Canada and became quickly integrated with the deaf community at local, provincial, and national levels.

## 8. Findings

### 8.1. Fatima

Fatima was interviewed by the research team about her dinner table experience. She reported that her family has become tolerant of sign language within her immediate circle of deaf friends, but does not use it with her. Her parents are not interested in learning sign language and use spoken English or spoken Bengali with her at the dinner table. She indicated feelings of boredom, isolation, and depression when being made to attend large family gatherings, as she is unable to keep up with the quick conversation patter between her immediate family, cousins, and older extended family members. When asked how she copes with the dinner table experience, she indicated that she resists through subversive tactics. For instance, she would tell her parents that she needed to use the bathroom and spend an extended period in the bathroom, texting her deaf friends and chatting with them before returning to the dinner table. Upon her return to the dinner table, her family members inquired after her health, wondering why she had been gone so long. She would reassure them by saying "Don't worry, I feel fine". Her response was to tell her family that she was chatting with other family members. She identified this as a form of trickery, an act of self-preservation, and a way to cope with the demands of the dinner table.

### 8.2. Kainat

Kainat learned sign language for the first time beginning at the age of 17. She can speak Urdu but has difficulty engaging in conversations with Urdu speakers. In her interview, she spoke of her younger deaf sister who learned sign language at a nearby elementary deaf and hard of hearing resource room within the same year upon arrival in Canada. In Pakistan, Kainat struggled in school with spoken and written forms of Urdu and English. She and her deaf sister began to use sign language with each other after having learned it at the Canadian school. When their deaf friends visit, they are forbidden to engage in signed conversations with each other. When asked about the dinner table, Kainat reported that when she attempted to converse with her sister in sign language at the table and in other spaces in the home, their parents would interrupt them and tell them to speak only in English. For this reason, Kainat reported that "I only talk to my sister in the bedroom". Kainat and her sister no longer converse with each other or with their family members at the dinner table. The use of sign language is forbidden in their home and they converse in private spaces such as their shared bedroom, away from the eyes of other family members and their parents.

### 8.3. Mustafa

Mustafa reports having been exposed to a nuanced and robust sign language for the first time in Canada. In contrast, the sign language used in his country of origin was delivered by unskilled teachers without the support of a deaf community who could provide native models of sign language. "It was mostly gestural, just pointing at things and acting out states of want, need, and emotions", Mustafa says. He could never learn to speak, and his parents primarily used gestures to communicate with him in the home. Upon learning ASL, he was eager to teach his family how to sign but his family indicated that they had become accustomed to this gestural system of homemade signs and were not motivated to learn a more intricate form of sign language complete with syntax, phonological and morphological rules, and expanded vocabulary. Upon arrival in a refugee camp in Syria,

efforts were made to procure him hearing aids at the age of 14. He wore them for a time but did not find them helpful in terms of learning to listen and speak. His parents urged him to wear the hearing aids in Canada, but he discarded these aids as they were of little use to him. Upon arrival in Canada, he was amazed at how much information could be conveyed through ASL in the context of a community of fluent ASL signers, and therefore embraced it wholeheartedly.

Mustafa's account of dinner table experiences with family and friends focused on not being able to access conversations through this rudimentary gestural system and being told that he would receive a full or coherent explanation for the conversations, arguments, and topics later, which never came about because of the gestural nature and limitations of their signing. He recalls, at one time, individuals making heated comments about a paper, to which he repeatedly asked for an explanation as to what was going on. After being told to wait, he took out his own letter (albeit an entirely different one) from his own pocket to study its contents. Immediately, the members of the dinner table demanded to know what the paper was about and snatched it from his hands, demanding to know where he had obtained this paper. In the interview, Mustafa explained that it was not fair that they snatched the paper away from him, demanding to know what was in the letter when moments before, he had asked for clarification on another letter being discussed by his family and friends.

### 9. Piece de Resistance: Madcap Misadventures of Mustafa

In the spring of 2022, the Deaf Crows Collective commissioned Kainat, Fatima, and Mustafa, with the support of Mooky McGuinty, to create several clown scenes that would precede the clown scene originally created and performed as part of a performance called *Apple Time* (Deaf Crows Collective 2018). In line with the shape-shifting nature of arts-based action research, these new pieces were to be added to the original piece and performed as one entirely new performance. The name of the entire clown show is *Madcap Misadventures of Mustafa*, and it premiered on 17 June and played on 18 and 19 June 2022. One of the new scenes is a dinner table scene which was performed six months after our research interviews.

This scene portrays Mustafa sitting at the dinner table between two women (hearing clowns) portrayed by Kainat and Fatima. Kainat is the clown mother in this scene and has created a delectable meal for her son and her husband. Mustafa is dragged reluctantly to the table but then becomes very interested in the food which he gobbles down with gusto. He makes several chewing noises and burps loudly at which the two women clowns titter politely, throwing deprecating looks at each other. Then, the clown father opens an Arabic newspaper and through a series of mouth movements and gestures, indicates that something scandalous and horrible has happened. Mustafa picks up on this exchange and demands to know what is in the newspaper. The clown father starts to explain to him and then realises that he cannot communicate with him and hands him a drink instead, encouraging him to drink. Meanwhile, the clown mother's "horror" escalates, and Mustafa becomes very alarmed and shakes his mother's arm, wanting to know what is truly happening.

The image (Figure 1) shows three actors on a stage wearing brightly coloured clothing. All wear red clown noses on their faces. The first actor has a long ponytail and wears a costume beard, and is looking at a newspaper with wide eyes, as if in a state of dismay, horror, and disgust at something in the paper. The second actor and the third actor are interacting with one another. The second actor, wearing a red and white striped shirt, is gesturing at something outside of the frame and looking at the third actor, shrugging with their other arm. The third actor stands on a chair with their hands on their head, as if in surprise or shock.

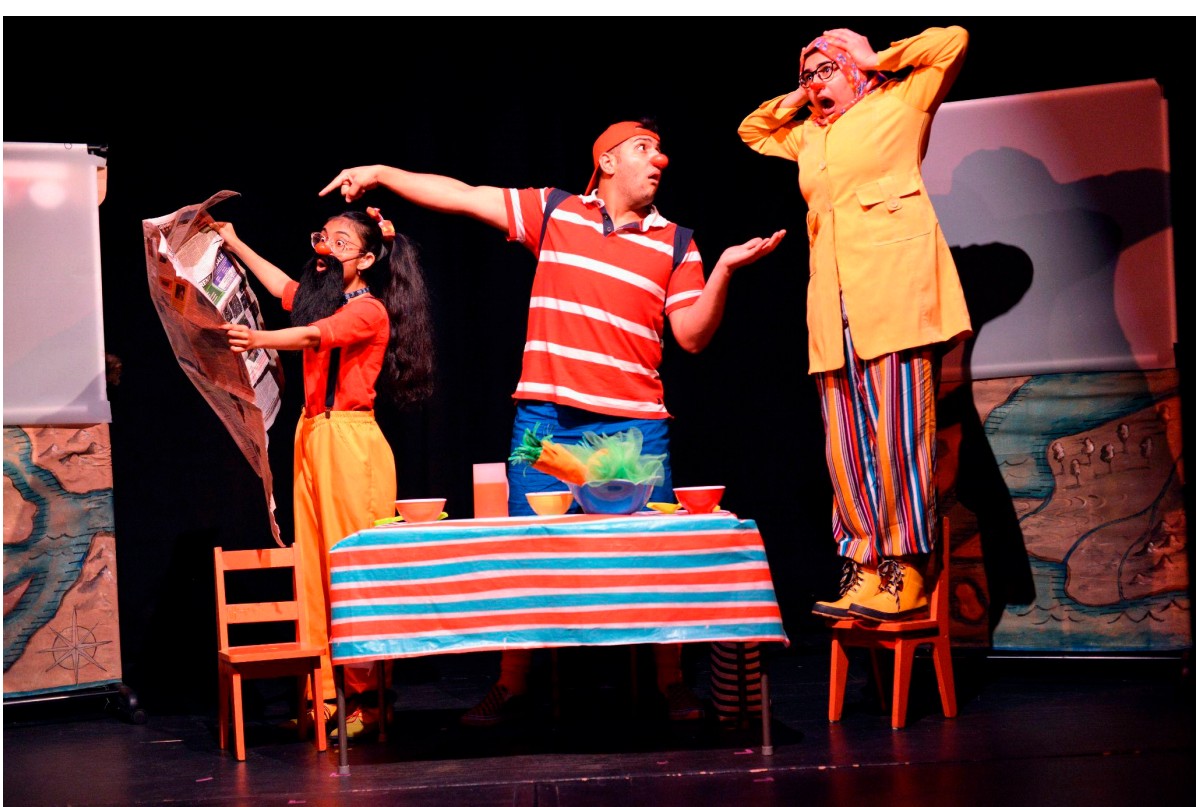

**Figure 1.** Dinner Table Scene in Madcap Misadventures of Mustafa (Deaf Crows Collective 2022).

The clown mother gestures at the dirty dishes on the dinner table, indicating that Mustafa should clean up the dishes. Then the two women continue their escalation, expressing dismay, horror, and disgust at something in the paper. Mustafa's clown mother holds one end of the paper while the clown father continues to hold the other end as they consult the paper. They slowly move away from the table, sweeping the paper over his head, leaving him with the dirty dishes on the table. Mustafa sadly picks up the dirty dishes, and pauses to gaze at the audience, evoking a sense of isolation, loneliness, and confusion.

## 10. Reverberations throughout the Deaf Youth Community: *From Deaf Shame to Deaf Same*

Mustafa, Kainat, and Fatima maintained their connections with younger and hard of hearing students through the activities of the Deaf Crows Collective. They engaged in several conversations with the younger deaf students during their time in the program and after they left high school. The collective partnered with an artist in residence, Chrystene Ells, and Regina Public Schools to create an exhibit called *From Deaf Shame to Deaf Same*. The younger students created an art display of the dinner table. The arts installation ran from 25 May to 21 July 2022 at the George Bothwell Library in Regina, Saskatchewan (Bamford 2022).

A photo (Figure 2) of an art installation. The installation includes a diorama wherein wooden figures of different colours (red, green, pink, blue, and orange) sit around a small dinner table in a suitcase with a painted family portrait above. The wooden figures are uniform in shape except for one family member, who is portrayed as a cut-out black-and-white photograph. Each figure faces a plate of spaghetti. In the portrait above, five blank-faced figures with coloured tops (red, green, pink, blue, and orange) are assembled, with one black and white figure painted in the foreground.

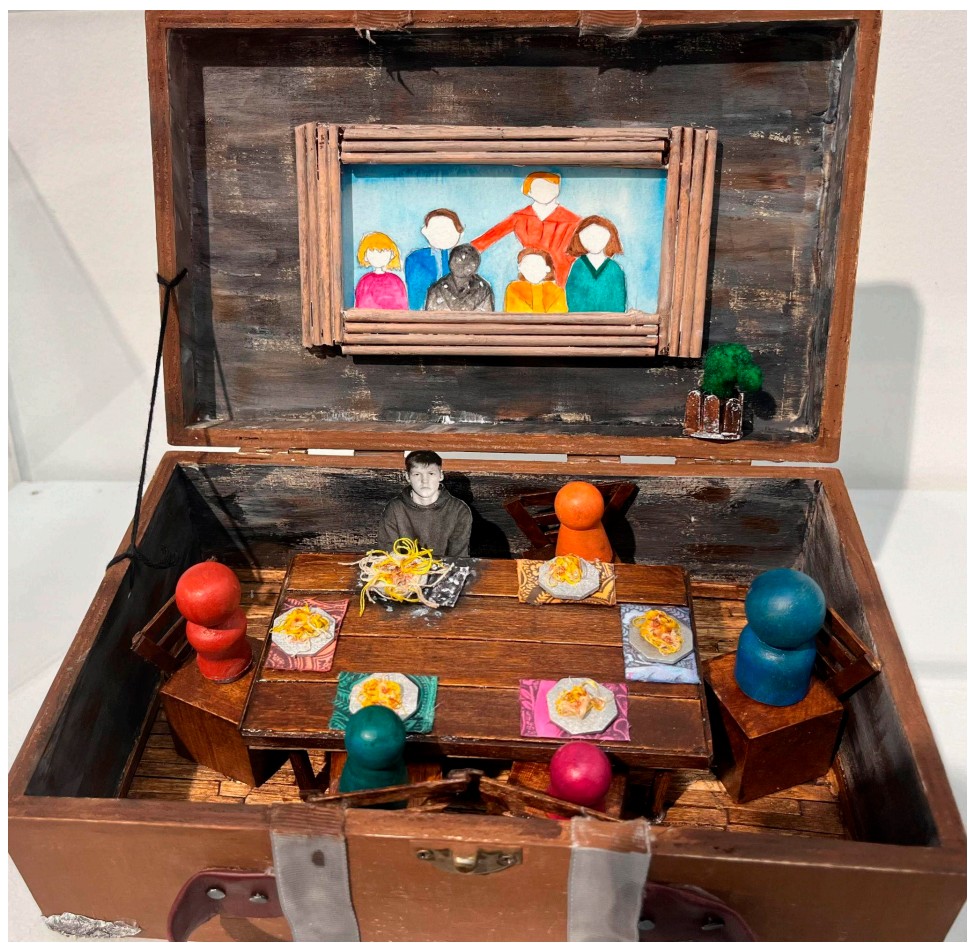

**Figure 2.** From Deaf Shame to Deaf Same (Winston Knoll Collegiate & Deaf Crows Collective 2022).

At this table, the deaf white student is portrayed in black and white, while the family members are featureless, but nevertheless brightly coloured. The family portrait above the table is portrayed in a similar fashion, with the deaf child greyed out and devoid of color. The table setting featuring a spaghetti meal shows that the spaghetti plates of the colourful family members are arranged inside the parameters of the plate. The deaf child's plate, however, shows the spaghetti escaping the borders of the plate and spilling out onto the table in an untidy fashion. The chairs upon which the colourful family members sit are turned toward each other and away from the deaf child, who is sitting in isolation.

## 11. Discussion

### 11.1. Investments in Identity

In considering our first research question (How invested are hearing and deaf interlocutors in their present and imagined identities? In what ways are they positioned by others, and how do they, in turn, position interlocutors in ways that grant or refuse power? How can hearing and deaf interlocutors gain from or resist these positions?), we found that the performers quickly embraced a deaf identity which empowered them to think about themselves and the world differently, including through lenses that considered race, gender, and positionings as young people in the composition of their own stories. Furthermore, their newfound deaf identities enabled them to work together to further their quest of becoming integrated into the world. Multiple intersecting identities also exact shifting and uneven flows of power. Fatima, Kainat, and Mustafa, being invested in their newly formed deaf identities, all expressed the desire for their parents and family members to embrace them as deaf people with deaf identities.

Nevertheless, parents and family members who are hearing continue to not sign at the dinner table with their deaf family members. Hall et al. (2018), in their analysis of the dinner table phenomenon, suggest that parental hearing status is now identified as a social determinant of deaf population health. Hearing parents may not see themselves as having a hearing "identity" or having audiocentric privilege in the same way that white people in a predominantly white society may not see themselves as having a white identity or privilege. Despite the performers' repeated efforts to perform stories on the stage and in film that point to their newly established deaf identities, their parents continued to explicitly emphasize spoken language in their homes. Although the deaf youth exhibit the cultural values of their families in dress, manner, and beliefs, and that of their new adopted country (Canada), they were unable to present these multiple identities at the dinner table. Rather, due to the emphasis on spoken language, they became persons without a history of relatable and shareable personal life experiences (Graif 2018) in every audiocentric culture including that of the home country and in Canada at the dinner table. Being stripped of a history, stories, and contexts which they can bring to the dinner table, they must somehow constrain and reduce themselves to what is expected of them. They are often considered as nonpersons because they are not able to hear and speak fluently in their home country's language or in English.

At the dinner table, expected experiences of translanguaging become difficult as these young adults must interiorly confiscate a deaf identity which feels palpable only to them and acquiesce to a state of nonbeing (Graif 2018), or worse, not being fully integrated members of their families, their home countries, or Canada. In this way, they are stripped of their multiple identities and must conform to the identities presented by their parents and families rather than using all their language resources as an expected outcome of family engagement across linguistic repertoires (Graif 2018).

*11.2. Investments in Cultural, Social, and Linguistic Capital*

Following Darvin and Norton's (2015) concern with structure and agency as it emerges at the intersection of identity, ideology, and capital, we think through these data in ways that attend to how translanguaging between deaf and hearing interlocutors at the dinner table is shaped with respect to this intersection. In doing so, we address our second research question: *What do nondeaf and deaf interlocutors perceive as benefits of investment, and how can the capital they possess serve as an affordance for learning?*

With their intersectional identity descriptions already established, we note that the families and extended families and friends of Fatima, Kainat, and Mustafa seemed uninterested in the newly discovered forms of capital held by their deaf family members. While Mustafa's family continued to use their gestures and some Arabic signs at the dinner table, they remained uninvolved in learning his new language. Fatima's family acknowledged that she used sign language, but they did not learn to use sign language with her at the dinner table. Kainat's parents outright forbade the use of sign language at the dinner table. The disjunction between their newly formed linguistic, cultural, and social capital and the forms of capital valued by their parents and family members may suggest acculturative stress often experienced by deaf members of hearing families because of differing languages, norms, and cultural values between deaf and hearing cultures (Aldalur et al. 2021). The expectation that these deaf youth listen and speak at the dinner table may be attributed to their positioning as young people in their families, meaning that they are on the lower end of ageist hierarchies that privilege adult knowledge and leadership in the domestic sphere. This expectation to behave in normative ways may also be described as a form of denial and therefore an act of marginalization (Aldalur et al. 2021). These expectations are outward forms of the investment in the belief that a deaf child can learn to speak and hear, a belief that is dominant and reinforced by medical approaches to hearing loss (Mauldin 2016). This unfortunate reality is often compounded by the long and protracted battle for the legal recognition of sign languages. Such debates about the validation of sign languages include consideration of language ideologies, public policy, and discourses about sign languages in

deaf communities (De Meulder et al. 2019b). With respect to the sometimes-invisible labour of investing in language learning described by Darvin and Norton (2015), O'Brien (2021) observes that "when a student or family attempts to campaign for language access or for recognition of their linguistic capital in sign language, they are not just fighting against a single teacher or school, but the whole educational establishment and the weight of history" (p. 72).

*11.3. Artistry as an Intervention on Barriers to Acquiring Capital*

The public art created by participants in this project speaks to our second question about the benefits of investment in multiple, intersecting identities. Given that translanguaging supports plurilingual encounters (Flores and Garcia 2017) and problem solving (Swain and Watanabe 2013), the participants' artistry offers significant insights about the significance of artistry in deepening our understanding of the complexities of translanguaging.

For the performers interviewed for this research, identity, ideology, and capital collide to create a culture of low expectations that is showcased at the dinner table in the *Madcap Misadventures of Mustafa* performance, where Mustafa's poor table manners and noises may be tolerated by women who throw disparaging looks at each other about his behaviour but do not correct him. The take-away for audiences is that even in his positioning as male in a scene that demonstrates patriarchal domestic dynamics, for Mustafa, being neither fully Syrian nor fully hearing seems to promote the view that he is also not fully human. Therefore, he becomes exempt from human social mores and behaviours. The same theme is echoed in the *From Deaf Shame to Deaf Same* exhibit in the family portrait, in which the deaf person is depicted as a black-and-white figure surrounded by family members who are wearing colourful clothes and whose skin colour resembles that of real human beings. The messy dinner plate juxtapositioned with the neat piles of spaghetti on the other family members' plates also supports the culture of low expectations. The family portrait suggests that humanity is not fully ceded to the deaf person by family members. Due to the availability of cultural, social, and linguistic capital shared by the deaf community, the deaf youth performers invested in an identity that remained largely invisible to their parents, and ironically was recognized by several hearing Canadians in the arts, academic, education, and social services sectors who invited them to give media appearances about their circumstances. Ironically, this identity investment that teetered between human and nonhuman categories and the capital granted by these categories earned them an unprecedented personhood and opened the door for translanguaging opportunities with other signing (hearing and deaf) people.

By contrast, engaging in the intertwined processes of interviews and arts-based action with Mustafa, Fatima, and Kainat allowed them to share the identarian and ideological experience of coming to Canada without recourse to positive role models in the deaf community within their country of origin. All had identified themselves as disabled, tasked with the struggle to read, write, and speak in their own country's language and again, upon arrival in Canada, to read, write, and speak in the English language (Alabssi et al. 2022). Upon entrance into a high school deaf resource program, they learned ASL in the classroom with other deaf students and their teachers (Weber 2018, 2021a, 2021b). They were also introduced to role models in the deaf community. Their participation in the Deaf Crows Collective as artists and youth performers enabled them to forge ties with each other and the adult deaf community amid a cultural backdrop of androcentrism and anglocentricity (Weber 2018, 2021a, 2021b). In this context, they found a source of cultural, linguistic, and social capital that gave them a positive identity, a sense of belonging, and a language, enabling them to collaborate and create art installations and professional theatre and film performances (Weber 2018, 2021a, 2021b). This newfound capital enabled them to liaise with other artists and actors who are hearing (Campbell 2021), and to speak to theatre and film directors, actors, educators, university academics, festival organizers, and prominent people in the deaf and hearing communities. This newfound capital was made possible through the acquisition of ASL, membership in the deaf community and the Deaf

Crows Collective, a professional arts organization that promoted their work and garnered media attention from television, radio, and newspapers on a provincial and national scale (Deaf Crows Collective 2022).

The rewards of embracing the available social, cultural, and linguistic capital served to strengthen participants' investment in an identity that positioned them as powerful, as having agency, and as capable. This newfound identity also was bolstered by the rising popularity of deaf artists and actors in the media, particularly the recent Oscar award winners of a film called CODA, which has an all-deaf cast including an 1989 Oscar award winner Marlee Matlin, and Troy Kotsur, who won the 2022 Oscar award for best supporting actor (Harris 2022). The significance of this engagement with artistry and, concurrently, capital, speaks to the breadth and vibrancy of the arts in such a way that translanguaging may emerge beyond the dinner table.

## 12. Language Ideologies

Finally, we posed the following research question involving language ideologies: What systemic patterns of control (policies, codes, institutions) make it difficult to invest and acquire certain capital? How have prevailing ideologies structured learners' habitus and predisposed them to certain ways of thinking? Language ideologies are often bifurcated, positioning the value of a language against another language and thereby creating language hierarchies, ranked by economic and social power (Canagarajah 2013). Beyond the intermingling of several languages, multiple ideological tensions and inequities abound at the dinner table: sign language versus spoken language; the medicalization of deafness and the search for a cure and the cultural affirmation of deaf persons, their culture, and the need for sign language; and the need to supplant home languages with spoken English upon immigration to Canada. Strict hierarchies reinforce the binaries pertaining to vocabularies, rules, and grammar believed to govern languages. Such hierarchies convey affordances pertaining to power and belonging (Canagarajah 2013; Garcia et al. 2015). The investment in spoken language, English in particular, has profound repercussions for the deaf youth artists at the dinner table. Kainat reported that she and her sister eat in silence and do not participate in family conversations, which are mostly conducted in Urdu and English. They know that if they attempt to sign to each other, their parents will interrupt and tell them to speak in English. Such monitoring engenders silence on their part as they cannot converse comfortably in a spoken language. Kainat eats in silence and waits for an opportune moment to communicate with her sister without being seen. For this reason, using all available linguistic resources is a privilege not always available to the participants in this study. Kainat's form of resistance ironically contributes to the language ideology that sign language is undesirable and is only to be used in private spaces and in conjunction with other, spoken languages. Sign language is the language of the powerless. The parents may be seeking to ensure that their deaf children will not be rendered powerless through their use of sign language.

At Fatima's dinner table, Bengali is the family language and English is the newly adopted language. Fatima's inability to converse comfortably in either language is met with indifference for the most part. However, Fatima resists this indifference by seeking out alternate spaces apart from the dinner table, such as by expressing the need to go to the bathroom and taking her phone with her so that she can call her deaf friends. Like Kainat's dinner table experience, in Fatima's home, two spoken languages dominate the dinner table and sign language is delegated to private spaces away from her family's eyes. This form of resistance keeps the languages apart and binarized in order of importance: Bengali, English, and last, sign language. For this reason, despite her resistance, sign language is also conducted away from "hearing" eyes. At the same time, the phone is a conduit to the outside world in which Fatima can sign with others. She uses this semiotic technology, which affords texts, video, and chat rooms, with deaf friends. She and her family members do not use the phone as a tool that could support translanguaging. For instance, voice-to-text, made possible through voice recognition software, is an option for

most phones. However, print literacy is difficult for many deaf students and is not the preferred modality for ease of communication. For Fatima, sign language continues to be marginalized at the dinner table.

Mustafa's family use some Arabic signs and agreed-upon gestures with Mustafa. Mustafa is largely fluent in ASL. He reports that ASL affords him depth of understanding, access to linguistic nuances, access to education, and participation in political debates. Mustafa's family can communicate with him effectively enough about basic topics using signs and gestures. In doing so, they may feel like they are using sign language but as Mustafa states, their signing appears to be more equivalent to a game of charades, in which information about complex social relationships and social history is absent. In other words, Mustafa's family may not be invested in learning ASL because they may think they know sign language without really knowing sign language (Graif 2018). This may be related to a commonly held ideology that sign language is not truly a language but a grouping of gestures that do not require attendance to syntactical, phonological, and morphological metalinguistic awareness (Kusters et al. 2020, Weber 2020).

This lack of investment seems to pose significant problems for Mustafa at the dinner table. Translanguaging is predicated on sustained access to languages, as experienced by hearing persons through listening and speaking, but this access appears to be cut off in the context of young, deaf newcomers' dinner table experiences. Translanguaging among deaf and hearing people, on the other hand, is always precarious (Snoddon and Weber 2021). Access to spoken language is not a given, nor are meanings quickly negotiated or obvious (Snoddon and Weber 2021). Hence, translanguaging is most effective when hearing persons possess a reasonable fluency in sign language, and does not entirely work to explain some of the above scenarios. We share the concern that "the use of multiple communicative tools is not necessarily something to be valorized in a sweeping movement, when it is an attempt by someone to create meaning from an impoverished set of linguistic tools" (De Meulder et al. 2019a, p. 10).

For example, the limited investment on the part of Mustafa's family has not evolved with Mustafa's acquisition of ASL. While there may be some opportunities for translanguaging at the dinner table, Mustafa often resorts to relying on simple gestures pointing to the "here and now of shared perception and memory" (Graif 2018, p. 23). In doing so, Mustafa must narrow his linguistic repertoire and deny the histories, complex social relationships, communities, and his multiple identities afforded by a rich and complex sign language to exhibit limited communication commonly associated with being a deaf Syrian Canadian who cannot speak (Graif 2018).

Darvin and Norton's (2015) model of investments supports a clearer explanation for translanguaging opportunities at the material and metaphorical dinner table for the young artists involved in this study. The families' expectation that their deaf youth were to conform to the norms, languages, and cultural values of the minority hearing culture and the dominant hearing culture, namely spoken and print English, resulted in the marginalization of the three deaf actors/artists at the dinner table. The families' investments in their own cultural, social, and linguistic capital and that afforded by Canadian society significantly diminished opportunities for translanguaging at the dinner table. At the dinner table with their family members, Fatima's and Kainat's translanguaging can be characterized as translanguaging restricted (Iturriaga and Young 2022) because they had to accommodate their parents' wishes that only spoken language in the form of English or their home country language was to be used at the dinner table. Fatima's and Kainat's limited spoken language skills, along with their parents' preference that their deaf family members use spoken language, seemed to fall within this category. In Mustafa's case, the family has come to accept that he cannot speak and resorts to gestures and signs to communicate with him. Mustafa reported using multiple semiotic resources such as props, utensils, paper, and movement to convey simple ideas, wishes, and requests at the dinner table, although this did not result in rich and nuanced conversation. Iturriaga and Young (2022) report this as expanded translanguaging because of his flexibility and creativity in

conveying his own ideas and the willingness of his family to use gestures and isolated signs. At the dinner table, however, his efforts at translanguaging were primarily restricted by the lack of access to spoken language (English and other languages). Mustafa became increasingly reliant on sign language interpreters to communicate about vital matters away from the dinner table. The types of translanguaging proposed by Iturriaga and Young (2022), however, do not fully describe a single communicative event such as the dinner table but provide a framework for understanding of the variety of translanguaging experiences.

Overall, undeterred by their families' contrary investments pertaining to cultural, linguistic, and social capital, language ideologies often reinforced by medicalized discourses about deaf people, and multiple identities, the youth performers continued to resist through art making, interviews in the media, and digital performances. Overall, the dinner table is a site of great tension that can only partially be explained through a translanguaging framework; translanguaging at the dinner table is inconsistent, emerging inequitably in some moments and not others. For example, when the young newcomers are met with indifference despite their initial protestations, they seek other avenues for self-expression, acknowledgement, and affirmation, such as through artistry that involves translanguaging in other contexts and through resistance in covert and overt ways that pushes back against normative expectations about language and communication established by their families: Fatima seeks out alternate spaces during the dinner table experience to communicate with her deaf friends; Kainat signs with her sister in the privacy of their bedroom; and Mustafa resorts to a patois of isolated Arabic signs, mime, and gestures to communicate with friends and hearing members at the dinner table. In their lives, the dinner table has become a metaphor for erasure and their silencing of their experience, but it is also the lever that promotes their art making, therefore opening up opportunities for translanguaging and other complex explorations of capital and power dynamics outside of and away from the dinner table.

## 13. Conclusions

The dinner table experience may be one of the most highly significant experiences wherein identity, language ideologies, investments in language learning, and cultural capital shape the translanguaging instinct and, therefore, translanguaging opportunities for deaf newcomer youth in Canada whose experiences at this literal and metaphorical setting demonstrate unresolved axiological conflicts that make the dinner table a space that cannot be explained through translanguaging alone. In this study, we sought to learn how the potential for translanguaging between deaf and hearing interlocutors at the dinner table is affected by the interrelationships between the hidden and known multiple investments in identity, ideology, and capital. Through interviews with these artists, paired with their public-facing artistic interventions, we demonstrate that translanguaging between deaf and hearing interlocutors at the dinner table is complex and predicated upon the interrelationships between the hidden, or "increasingly invisible", and known multiple investments with respect to capital, identity, and ideology available at and beyond the dinner table (Darvin and Norton 2015). As such, translanguaging is present in these participants' experiences but does not entirely or adequately explain what happens between three young, deaf, newcomer artists and their hearing family members. Ultimately, our findings reveal a chronic gap in translanguaging as demonstrated by participants' experiences; the investments of the interlocutors at the dinner table remained hidden and inexplicit, therefore muffling translanguaging potential. For this reason, we suggest that stakeholders that aim to support deaf newcomer youth and expand their linguistic and semiotic resources must do so in ways that include and stretch beyond translanguaging frameworks. Such stakeholders might include resettlement agencies, school programs, and counselling agencies that would benefit from raised awareness of these investments to facilitate translanguaging among deaf and hearing immigrants at the dinner table and in multiple other contexts in ways that support increased cultural, social, and linguistic capital. However, axiological commitments to understanding complex translanguaging

practices in the domestic sphere—or at the dinner table—also need to be examined in a way that includes but also extends beyond current investments in translanguaging.

**Author Contributions:** Conceptualization, J.C.W.; data collection, J.C.W. and A.A.; writing—original draft preparation, J.C.W.; writing—review and editing, C.T.J., A.A. and J.C.W. Funding acquisition, C.T.J. and J.C.W. All authors have read and agreed to the published version of the manuscript.

**Funding:** This research was funded through The Social Sciences and Humanities Research Council (SSHRC) Insight Development Grant titled, "Troubling Vocalities: Disability and Deaf Art on the Canadian Prairies" (430-2020-00189). Funding was also provided by The Council for Research in the Social Sciences (CRISS) of the Faculty of Social Sciences at Brock University (October 2020) and by the Social Justice Research Institute (SJRI) through an SJRI Community Engagement Grant (Beyond Niagara) at Brock University (336-242-071).

**Institutional Review Board Statement:** The study was conducted in accordance with the Declaration of Helsinki, and approved by the Institutional Review Board (or Ethics Committee) of Brock University (protocol code 20-261-JONES), approved 21 March 2021. University of Alberta, Pro00109652, approved 6 May 2021.

**Informed Consent Statement:** Informed consent was obtained from all subjects involved in the study.

**Data Availability Statement:** Not applicable.

**Conflicts of Interest:** The authors declare no conflict of interest.

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
