# Peer review of "Please Pass the Translanguaging: The Dinner Table Experience in the Lives of Newcomer Canadian Deaf Youth and Their Families"

_languages, doi:10.3390/languages8020096_

Round 1

Author Response

Thanks for your comments, see the details in the attachment.

Reviewer 2 Report

General comments:

Overall, the topic is exciting and relevant. The problem is emerging in the face of a growing trend of displaced populations, including deaf individuals in refugee or migrant situations.

In my view, the metaphor of the "dining table" constituted a creative and artivist motto to problematize an ecological situation of everyday life, with DHH people living with anxiety, and thus, deprived of a sense of belonging in home-making experiences, in cases in which sign language is not used due to ignorance or not. is permitted by parental conviction. the topic is very interdisciplinary in its implications. 

Focused comments: 

1. In the template proposed by the MDPI journal, ref. bibliographic in the abstract

2. lines 100 and 101 contain elements to improve: " community clinic settings" - I think there will be other relevant contexts; "successful incidences of translanguaging"- reports? experiences?

3. the concept of semiotics is not used clearly, namely when, related to translanguaging, it refers to linguistic resources to increase the communicative repertoire. They are not, therefore, objects, but systems of language, which visual language, scenic language, etc. (line 110)

4. lines 131-133

the main question appears late in the manuscript. The reader should be able to know it in the introduction

5. lines 199-201

text format to correct

6. Arts Based Action Research requires a careful presentation of "participants" in a specific chapter, When talking about 3 "newcomer Canadian deaf youth" artists in the intro, it's not immediately clear why we have 6 fictitious names in the findings. If that has to be with the  Deaf Crows Collective elements and artistic co-creation, this relation is not clear at a pertinent time in manuscript

7. line 288

what does this mean? "Research ethics for this study was obtained"

8. line 371-373 

gender clarification

9. references in text 

to be corrected  (XXXX, 2022, in press)

10. Art-based research

The Art-based research approach presupposes a level of intra and inter-personal transformation. The fact of introducing the term therapeutic here is not consensual- transformative effect, emancipatory, concept of agency - all seem to be stronger in the evidence than "healing" power, not doubting that it was involved, but in the text it seems out of place. ABR is little valued in the "conclusion" as a methodology and powerful tools to facilitate translanguaging among deaf and hearing immigrants, and even to replicate in language learning programs for plurilingual educational contexts.

10. Discussion (interviewed subjects/ participants?

In the discussion, how were the verbal data analyzed (NVivo)? and how was the content analysis of the performances and artistic objects carried out? Was it a semiotic analysis? How were the qualitative data from these two data collections (interviews and artistic production) crossed?

Author Response

(The authors gave the same response as above.)

Reviewer 3 Report

This manuscript explores the phenomenon of the dinner table experience, or dinner table syndrome, among deaf newcomer Canadians and the relationship of translanguaging to this experience. I believe that this type of research would be valuable for understanding the complex language use in these environments, but I have suggestions for how to improve the clarity of the. manuscript and the connection to translanguaging.

Introduction and literature review:

The opening paragraph to me had several points that could be expanded upon, specifically defining newcomer Canadians as they will be the focal point of the study. What is your technical definition for someone being a newcomer Canadian? Why are you interested in this population in particular, as opposed to deaf individuals in hearing families of other kinds of backgrounds? Understanding why the focus is on newcomer Canadians, and what we know about the experiences of newcomer Canadians, would be a valuable framing device. For instance, line 176 stated that the hidden and known investment particularly for newcomer Canadian families with deaf children – that is true for all hearing families with deaf children.

It would also be helpful to frame for the audience why the authors chose to use the translanguaging framework to explore dinner table syndrome? It seems that the goal is to use translanguaging to demonstrate the power dynamic among the language. That would require some critical stance to examine the imbalance power dynamic where translanguaging is used to capture to describe the actions during the dinner table that cause the syndrome for participants. The authors mentioned the Critical Translanguaging Space, so the reviewer feels there should be a section defining this phrase.

The syndrome is discussed at an individual level as a deaf family member in this paper. The author should also discussing the syndrome at the social level.

A note - Middle Eastern is a Eurocentric term that promotes the language of colonization. By re-framing our geographical understanding, the region could be called SWANA (Southwest Asia & North Africa), or the author could name a specific country.

The paragraph from lines 48-74 is very light on citations, especially from lines 56-72. There is enough research on semiotics and even gesture that could lend support to what is being described here.  The epistemological and ontological stance of the dinner table syndrome is also not fully described – would like to see more literature discussing the significance of dinner table conversation and how the quality of dinner conversation influence the individuals.

Methods:

What was the approach for translation of the interviews from ASL to English? Given that 2 of the 3 authors are not ASL proficient, I wonder about whether there was someone available to affirm the accuracy of the translation?  

Was there any kind of reliability measure undertaken with regard to the actual coding of the data?

What were the specific backgrounds of the three participants? A participants section [SDC1] with relevant information about them seems warranted. The participants are introduced more fully in the findings, rather than in the methods. It is hard to follow who are the artists before introducing the artists, same issue with the names of the productions the artists made.

For the passage beginning on line 222 – would be nice if the authors could describe art-based research before talking about their participants’ products. The writing felt abrupt with discussing the participants’ products without defining the art-based action research approach.

Results:

Line 333 – what is being referenced here? The authors seem to be describing a specific participant’s age at learning sign language, so the citation was surprising to see.

Lines 366-370 – this interaction is confusing. It seems that this was a letter – to Ahmad? To the family? To someone else? It’s not clear how Ahmad acquired it and why the family’s reaction was so harsh. Some clarification here would be helpful for understanding this incident.

I thought the inclusion of the arts pieces here was fascinating.

I expected to see the themes and codes that emerged from the data described in this section, but I did not encounter them. What were the themes? How did the similar themes manifest across the different individual experiences?

I was also a bit surprised that translanguaging was not explored in the results section – in fact, a search reveals that the word was not included anywhere in this section. How do the results reveal translanguaging in action? I would have expected this to potential show up in the codes or be explicitly called out in the description of their reported experiences. SC: Perhaps by defining the translanguaging at the introduction will help the authors capture or notice absence of translanguaging.

The results shared good findings about the participants, but it is written in chunks with unclear transitions from interview data to performer data to performance to production.

Discussion:

I was especially moved by the description of what is almost reverse translanguaging, the hearing families’ refusal to use any means of communication outside of their native, spoken language.

Overall, I felt the discussion was much more aligned with translanguaging explicitly, and well written. I would have wanted to see more of this existing explicitly in the results section so that the results and discussion could be clearly aligned. Lines 590-653 in my opinion could be moved to the results section as they are a clear discussion of the data reported in the interview connected explicitly with translanguaging. Perhaps the parts of this that deal explicitly with these data can be separated from the parts that connect with other research, which could/should remain in the discussion.

Some overall notes:

 The authors used many specific terms for the study but need to take care on explaining how these terms are contextualized in their study such as translanguaging, art-based research, critical translanguaging space, and other theoretical and epistemological frameworks to their readers. It can become confusing to infer these concepts, so the readers would appreciate explicit description of these conceptions in this study.

One of the notable limitations in this study is the data comes from the deaf participants’ experience during the dinner table that doesn’t take account of the perspectives of their family members. What is the parents’ perspective of dinner table conversation? How did they use translanguaging to foster or hinder dinner table conversation with their deaf children? That would be an interesting perspective to see in what ways translanguaging could become an asset or barrier to facilitate dinner table conversation between hearing and deaf family members.

Additional literature recommendation for translanguaging in deaf community:

-       Translanguaging & Identity: Napier, J., Oram, R., Young, A., & Skinner, R. (2019). “When I speak people look at me” British deaf signers use of bimodal translanguaging strategies and the representation of identities. Translation and Translanguaging in Multilingual Contexts, 5(2), 95-120.

-       Translanguaging & Family: Kusters, A., De Meulder, M., & Napier, J. (2021). Family language policy on holiday: four multilingual signing and speaking families travelling together. Journal of Multilingual and Multicultural Development, 42(8), 698-715.

-       Translanguaging and immigrant students: Allard, K. & Wedin, A. (2017). Translanguaging and Social Justice: The case of education for immigrants who are deaf or hard of hearing. In B. Paulsrud, J. Rosén, B. Straszer, & A. Wedin (Eds), New Perspectives on Translanguaging and Education, 90-107.

Author Response

(The authors gave the same response as above.)

Reviewer 4 Report

Dear authors,

Thanks for your paper, I enjoyed reading it! I think there’s potential for a very good contribution to the field here, but it needs a bit of work to bring out the best in it. I’m going to go through the paper and offer my comments in the order they arise. However, some are much more important than others. For example, the research ethics section really needs to be sorted out, because the way you present it potentially causes a pretty serious breach in ethics.

Firstly, I find the way the paper is structured a bit confusing in parts. For example, on p.2 you talk about devising a sophisticated multi-layered theatre performance, but you don’t offer any context as to how/why this performance was devised. Was it part of the research project? Was it a data collection exercise? You need more context here and explain exactly what exactly the relationship between the performance and your project. You do offer more detail on this later on, but it’s still not clear whether the performance was separate to the research or not. You really need to re-think how you introduce your data collection and participant involvement in the project.

The epistemology/ontology/axiology section is good, but it’s missing an explanation of exactly WHY the dinner table syndrome is so important. You need to draw out more here about what exclusions deaf people experience more widely (missing out on incidental learning in school, missing out on access to mainstream media etc. that a hearing child would be expected to use to pick up information) and then link it to the dinner table syndrome. Make it clearer it’s a level of exclusion that deaf people suffer on top of what else they experience outside the family home, and why this is contrary to what you would expect of the family practice of breaking bread together. Then you can move on to emphasise the particular situation of deaf newcomer Canadians and why the dinner table is particularly problematic for them. At the moment, the importance of this doesn’t really come across. Clarity on this point will help with your analysis later on because it will help the reader break down the different elements at play here.

Incidentally, I have never come across the term ‘newcomer Canadian’ before. I don’t think it’s in common circulation outside Canada/North America, so a footnote or sentence explaining what the term means would be useful for international readers.

In your methodology, and throughout, you need to explain more about the concept of capital that you use. There are many different interpretations of capital, and linguistic, cultural and social capital. Often these are very different to one another. Without you clarifying which interpretation you are using, it’s very difficult to say whether your analysis is sound or not. For example, if you use Bourdieu’s concepts, you can’t say that learning a language results in increased cultural capital. Bourdieu sees capital as a result of labour – you don’t simply ‘acquire’ capital, you have to work for it. Other theorists have a different view. You need to be much clearer here about what the terms you use actually mean, where you are taking them from, etc.

I don’t think that you actually address the 3 research questions that you present on p.5 in this paper. You talk about ‘present and imagined identities’ but don’t provide us with a theoretical framework through which to understand those terms. You talk about investment/capital again but we’re still not clear on what these terms mean. You talk about systemic patterns of control, but it’s not clear how these relate to the dinner table syndrome. I would recommend either explicitly addressing each of these RQs in your paper, or leaving them out.

I think the arts-based action research section is interesting but it doesn’t feel well integrated with the rest of the paper and the theoretical framework you’ve supplied. It’s not clear how this section relates to the rest of the paper as a whole. Again, were the artworks/performances created specifically for the research project, or did they just happen to be occurring at the same time? All the info about the performances etc. needs to be earlier on in the paper – it’s really important context to help us understand what you did and why. At the moment it’s a struggle to see what’s going on for the first 6 pages or so until we get to this point. Much better to move all this info much earlier in the paper to provide this essential context earlier on.

Were the researchers doing the interviews deaf or hearing? Were they fluent in ASL? Were they also newcomer Canadians? Important information which will help us understand the interview context.

The section on ethics is concerning. You claim you promised anonymity to all participants. But it’s very easy to compare the interview summaries with the performer data and figure out who is who. If you promised anonymity to the interview participants, you are breaking this promise by sharing performer information in such a way that it is easy to identify the interviewees. I think you also need to be clear on how you define confidentiality. If someone tells you something in confidence, you can’t share it. So how are you sharing interview data if you’d promised all information would be confidential?

If participants knew that they would be easily identified in this paper because of their performer data, you need to make this really clear that they knew and agreed to waive their right to anonymity. If they didn’t agree, then you need to find a different way of presenting the interview data which does not break your promise of anonymity. This is pretty much a make-or-break point for the paper. Unless this point is resolved I can’t recommend it’s published.

Presuming you resolve the above issue, you need to include more actual quotes from the interviews. You’re presenting summaries, but with no actual data to support them. Actually, you do have one short quote from Ahmad. But you need much more than this. Without data, we don’t know whether your summaries are accurate or not!

On page 12 again there’s the issue of using the terms ‘capital, cultural etc.’ without defining what they actually mean.

There are a few things on p.13 which are not clear. On lines 551, 552, you use the word ‘their’, but it’s not clear to who you are referring. On lines 562-563 you talk about ‘interiorly confiscating’ a deaf identity. This is an intriguing turn of phrase, but I’m not clear what this means – could you expand on this?

The section on translanguaging is interesting, and I think if you make the clarifications/elaborations on dinner table syndrome I suggest above you will be able to make much more of this. You will be able to compare the dinner table experiences of newcomer Canadians to those of other deaf people who don’t have the extra layers of culture and language that your participants do. You’ll be able to expand on this section with more clarity than you have at the moment.

Overall, there’s a lot of potential here. The most important issue, of course, is the ethical one. With additional structural changes and clarifications as mentioned above, I think this could be a strong paper.

Author Response

(The authors gave the same response as above.)

Round 2

Reviewer 1 Report

Bee attachment.

Author Response

Thanks for you comments and see the responses in the attachment

Reviewer 3 Report

I would like to thank the authors for their revisions to this manuscript. I believe the connection to the concept of translanguaging is more robust, and the sections of the manuscript feel more coherent now. I have only a couple of minor suggestions that I hope the author will consider. 

-When describing signed language, the authors often use the term "sign" alone, which can be a bit ambiguous. I personally prefer to use terms like "sign language," "signed words," etc to be a bit more specific. This could also disambiguate from other signed systems, like signed English, which could also be described as "sign"

-It is my understanding that instead of "minority" the current preferred term is "minoritized." I believe this term only occurs once in the manuscript, and I would recommend changing it to follow the preferred term.

Author Response

Thanks for your comments.

Reviewer 4 Report

Thanks for this thoughtful engagement with the feedback given. This is a really good paper now which I very much enjoyed reading! Thank you!

Author Response

Thanks for your comments.